# Extremely broadband, on-chip optical nonreciprocity enabled by mimicking nonlinear anti-adiabatic quantum jumps near exceptional points

Youngsun Choi[1], Choloong Hahn[1], Jae Woong Yoon[1], Seok Ho Song[1] & Pierre Berini[2,3,4]

Time-asymmetric state-evolution properties while encircling an exceptional point are presently of great interest in search of new principles for controlling atomic and optical systems. Here, we show that encircling-an-exceptional-point interactions that are essentially reciprocal in the linear interaction regime make a plausible nonlinear integrated optical device architecture highly nonreciprocal over an extremely broad spectrum. In the proposed strategy, we describe an experimentally realizable coupled-waveguide structure that supports an encircling-an-exceptional-point parametric evolution under the influence of a gain saturation nonlinearity. Using an intuitive time-dependent Hamiltonian and rigorous numerical computations, we demonstrate strictly nonreciprocal optical transmission with a forward-to-backward transmission ratio exceeding 10 dB and high forward transmission efficiency (∼100%) persisting over an extremely broad bandwidth approaching 100 THz. This predicted performance strongly encourages experimental realization of the proposed concept to establish a practical on-chip optical nonreciprocal element for ultra-short laser pulses and broadband high-density optical signal processing.

[1] Department of Physics, Hanyang University, Seoul 04763, South Korea. [2] School of Electrical Engineering and Computer Science, University of Ottawa, 800 King Edward Avenue, Ottawa, Ontario, Canada K1N 6N5. [3] Department of Physics, University of Ottawa, 150 Louis Pasteur, Ottawa, Ontario, Canada K1N 6N5. [4] Centre for Research in Photonics, University of Ottawa, 25 Templeton Street, Ottawa, Ontario, Canada K1N 6N5. Correspondence and requests for materials should be addressed to J.W.Y. (email: jaeong.yoon@gmail.com) or to S.H.S. (email: shsong@hanyang.ac.kr).

Nonreciprocal light propagation is the key performance attribute of optical isolators and circulators, elements essential for optical signal processing, telecommunications and the protection of high-power laser systems. Whereas the Faraday effect in magneto-optic crystals enables broadband and high-efficiency nonreciprocal elements for free-space systems, the realization of integrated on-chip nonreciprocal elements remains elusive. Different approaches have been proposed to realize integration-compatible, on-chip nonreciprocal devices. To this end, indirect photonic transitions mediated by dynamic index modulation[1,2] and nonlinear resonance shifts in asymmetric high-Q microcavities[3–5] have been studied in depth. These effects produce remarkable nonreciprocal transmission within a sub-mm or even µm-scale device footprint; however, other performance attributes are problematic such as a narrow bandwidth, low forward-transmission efficiency and a high signal power threshold for operation. These problems must be eliminated or at least significantly alleviated to process broadband signals (including modulation) to respect tight energy-conservation constraints imposed by applications and for stable operation in ambient conditions involving moderate temperature drifts.

Optical nonreciprocity in nonlinear parity-time (PT) symmetric systems[6–8] are presently attracting considerable attention. In PT-symmetric coupled microcavities, optical modes undergo a spontaneous symmetry-breaking transition at an exceptional point (EP). Entering further into the broken-symmetry phase leads to enhanced cavity-excitation asymmetry between two opposite coupling directions. Including auxiliary nonlinear effects, such as the optical Kerr effect and gain saturation[6,7], strong nonreciprocal transmission is obtained as opposed to strictly reciprocal PT-symmetric effects in the linear and stationary systems[9–16]. In spite of high nonreciprocal transmission ratios and low signal power thresholds for operation, the functionality of such approaches is available only over an ultra-narrow bandwidth on the order of MHz and involves unpredictable laser oscillations causing critical instability. These problems are difficult to avoid in high-Q resonator-based approaches.

In this paper, we propose an integration-compatible, nonresonant broadband nonreciprocal device concept inspired by non-Hermitian quantum-mechanical interactions near an EP. We employ a time-varying non-Hermitian Hamiltonian along a parametric path that encircles an EP where the canonical quantum adiabatic theorem breaks down exclusively for one preferred temporal direction. A spatial analogy of this effect is manifested in a nonlinear coupled-waveguide structure with amplifying and attenuating waveguides. Importantly, the obtained asymmetric optical propagation is totally unrelated to interferometric power beating or any resonant optical excitations that result in a strong wavelength dependence. Thus, insensitivity to the excitation wavelength is obtained as opposed to previously proposed schemes. Including normal gain saturation and the consequent power regulation effect yields robust nonreciprocal transmission over an extremely broad spectral band that is only limited by the gain bandwidth of the gain material selected.

## Results

**Design and basic performance.** We consider a waveguide architecture shown in Fig. 1a. The system comprises a unidirectional mode converter section implemented as nonlinear coupled waveguides connected to input and output Y-branches and single-mode input and output waveguides. The unidirectional converter is a key functional region where one-way adiabatic modal transformation occurs. For forward (left-to-right) propagation, an incident mode from the input single-mode waveguide

and Y-branch preserves its transversal symmetry during propagation over the converter region, and thus freely transmits to the output waveguide. For backward (right-to-left) propagation, an incident symmetric mode is converted into an antisymmetric output mode that is eventually rejected at the Y-branch because of modal incompatibility with the single-mode waveguide.

To produce the envisioned nonreciprocal transmission, we apply complex effective index profiles in the coupled-waveguide section, as indicated in Fig. 1b. The effective indices $n_1$ and $n_2$ of the fundamental guided modes in the upper and lower arms, respectively, are defined as $n_p = \beta_p/k_0 = n_c + \Delta n_p(z)$, where $p$ takes on 1 for the upper attenuating waveguide or 2 for the lower amplifying waveguide, $\beta_p$ is the propagation constant of the fundamental guided mode in waveguide $p$, $k_0$ is the vacuum wavenumber and $n_c$ is the average real effective index. Here, $\Delta n_p(z)$ is the desired complex modulation profile given by

$$\Delta n_1(z) = \Delta n_0 \left\{ \sin\left(\frac{2\pi z}{L}\right) + i\left[1 - \cos\left(\frac{2\pi z}{L}\right)\right] \right\}, \quad (1)$$

$$\Delta n_2(z) = -\Delta n_0 \left\{ \sin\left(\frac{2\pi z}{L}\right) + iS(I_2)\left[1 - \cos\left(\frac{2\pi z}{L}\right)\right] \right\}, \quad (2)$$

where $L$ is length of the converter region and $S(I_2) = (1 + I_2/I_S)^{-1}$ is a gain saturation factor with saturation intensity constant $I_S$ and local intensity $I_2$ in waveguide 2. We note that the specific $\Delta n_p(z)$ profiles described by equations (1) and (2) are a representative case among a wide variety of other possible profiles, as we will explain in the next section.

Figure 1c–e summarizes two-dimensional finite element method calculations assuming single-mode slab waveguides with $\Delta n_0 = 4.26 \times 10^{-4}$, $L = 5$ mm, a waveguide core width of 1 µm, a core separation distance of 2 µm and an operating wavelength of 1.13 µm (other parameters and conditions are given in the figure caption). For an isolated waveguide of these parameters, the maximum linear modal gain/attenuation constants are $\pm 411.4$ dB cm$^{-1}$ and the total amplification/attenuation over the 5-mm-long unidirectional converter region are $\pm 102.8$ dB. These levels of optical gain and loss are readily obtainable with a variety of optical gain materials such as conventional direct bandgap semiconductors and dye-doped polymers. We observe in Fig. 1c,d that forward propagation yields significantly amplified transmission at the right-hand single-mode waveguide output, whereas most of the output mode energy diverges into the cladding for backward propagation, thereby transmitting a negligibly low optical power. The output intensity profiles in Fig. 1e for the forward and backward propagation cases clearly demonstrate high-quality nonreciprocal transmission ratio (NTR) of 28.7 dB and amplified forward transmission with a gain of $+4.95$ dB. This nonreciprocal effect in a stationary and nonresonant system is caused by a nonlinear non-Hermitian wave interaction near a PT-symmetric EP as we will explain in the next section.

**Principle and fundamental properties.** We describe the dynamics of optical modes over the converter region using the coupled-mode formalism:

$$\frac{d}{k_0 dz}\begin{bmatrix} A_1(z) \\ A_2(z) \end{bmatrix} = i\begin{bmatrix} \Delta n_1(z) & \kappa \\ \kappa & \Delta n_2(z) \end{bmatrix}\begin{bmatrix} A_1(z) \\ A_2(z) \end{bmatrix}, \quad (3)$$

where $\kappa$ is the coupling constant and $A_p$ denotes the amplitude of the fundamental mode in waveguide $p$. We write the frequency-domain total electric field of the coupled section as $\mathbf{E}(x,y,z) = [A_1(z)\mathbf{E}_1(x,y) + A_2(z)\mathbf{E}_2(x,y)] \cdot \exp(in_c k_0 z)$ with $\mathbf{E}_p(x,y)$ indicating the normalized wavefunction of the fundamental mode in waveguide $p$. In the low-intensity limit where the gain saturation

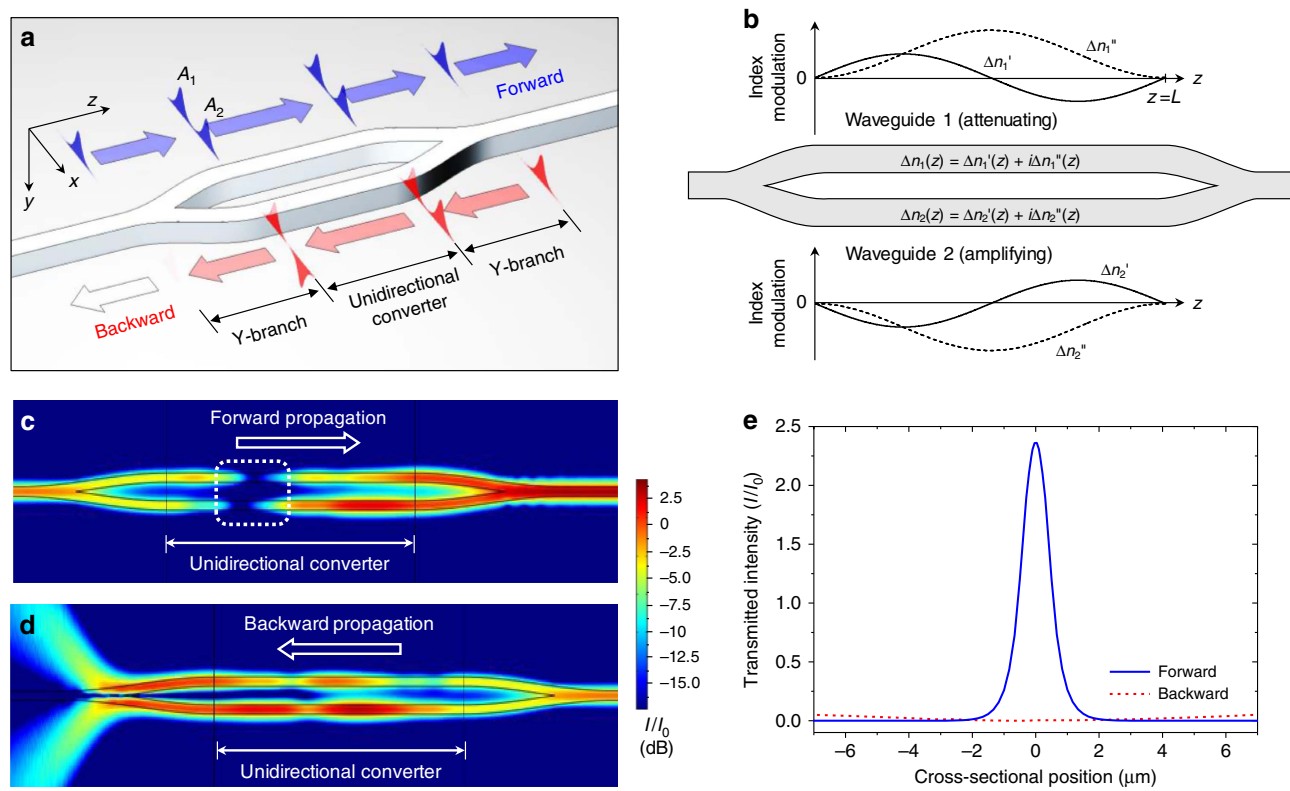

**Figure 1 | On-chip optical nonreciprocity based on nonlinear dynamics near an exceptional point.** (**a**) Basic operating scheme of the proposed concept. (**b**) Effective index modulation profiles over the unidirectional converter region for the attenuating (upper) and amplifying (lower) waveguides. (**c,d**) Two-dimensional numerical calculations (FEM, Comsol Multiphysics) assuming single-mode slab waveguides with the following optical parameters: cladding index of 1.50; core index of 1.55; core index modulation depth $\Delta n_{core} = 6 \times 10^{-4}$. The effective index modulation amplitude is given by $\Delta n_0 = \Gamma \Delta n_{core}$ and the confinement factor is $\Gamma = 0.71$. Operation at the vacuum wavelength of 1,130 nm and in the fundamental transverse-electric mode ($TE_0$) is assumed. The colour scale indicates intensity $I$ in dB with respect to the incident intensity $I_0$. The peak intensity of the incident mode is $I_0 = I_S$ for both the forward and backward transmission cases, where $I_S$ denotes a gain saturation intensity constant. (**e**) Cross-sectional profiles of the transmitted intensity in the forward and backward directions. The estimated nonreciprocal transmission ratio (NTR) is 28.7 dB and the forward transmission gain is 4.95 dB.

factor $S(I_2) \approx 1$, equation (3) can be expressed as a linear Schrödinger-type equation $d|\psi_{fw}(t)\rangle / dt = i\mathbf{H}(t)|\psi_{fw}(t)\rangle$ with an effective Hamiltonian

$$\mathbf{H}(t) = \begin{bmatrix} \xi(t) & 1 \\ 1 & -\xi(t) \end{bmatrix}. \qquad (4)$$

where we define the forward dynamic state vector $|\psi_{fw}(t)\rangle \equiv [A_1(t) \ A_2(t)]^T$, the fictitious time variable $t \equiv \kappa k_0 z$ and the reduced energy parameter $\xi(t) \equiv \Delta n_1(t/\kappa k_0)/\kappa$. The parametric spectra of eigenvalues $\lambda_\pm = \pm(1+\xi^2)^{1/2}$ of $\mathbf{H}$ on the complex $\xi$ plane have characteristic singularities at $\xi = \pm i$ corresponding to a pair of PT-symmetric EPs. A comprehensive description and the general features of non-Hermitian singularities of this kind are found in ref. 17. On the complex $\xi$ plane, the effective index modulation profiles given by equations (1) and (2) imply circular trajectories around the EP for $\Delta n_0 > \kappa/2$, as shown in Fig. 2a. For the simulated case in Fig. 1c–e, $\xi(t)$ encircles the EP at $\xi = +i$ counterclockwise. The backward propagation is described by a time-reversal transformation $t \to T-t$ and $\xi(T-t)$ encircles the EP clockwise, where $T = \kappa k_0 L$ is total (fictitious) time duration for a single parametric revolution. Time evolution of the backward dynamic state $|\psi_{bw}(t)\rangle$ is thus governed by the time-reversed Schrödinger-type equation $d|\psi_{bw}(t)\rangle / dt = -i\mathbf{H}(T-t)|\psi_{bw}(t)\rangle$.

For our time-varying Hamiltonian $\mathbf{H}(t)$, the instantaneous eigensystem, determined by a local eigenvalue equation $\mathbf{H}(t)|\phi_\mu(t)\rangle = \lambda_\mu(t)|\phi_\mu(t)\rangle$, reveals a typical complex square root

distribution as plotted in Fig. 2b. Rigorously defining the instantaneous eigenvalue–eigenvector pair $\{\lambda_\mu(t), |\phi_\mu(t)\rangle\}$ such that they are continuous functions of $t$ for $0 \le t \le T$, we use a branch cut at $\text{Re}(\xi) = 0$ for $-1 \le \text{Im}(\xi) \le 1$. We show $\lambda_\mu$ surfaces on the complex $\xi$ plane and two trajectories for $\lambda_\mu(t)$ in Fig. 2b. The eigenvalue surface is divided into two sheets by the branch-cut demarcation. A sheet representing $\lambda_G$ ($\lambda_G$ sheet) has a negative imaginary part as indicated by the white–blue–cyan skin and the other sheet for $\lambda_L$ ($\lambda_L$ sheet) has a positive imaginary part as indicated by the white–red–yellow skin. For $\xi(t)$ corresponding to equations (1) and (2), $\lambda_G(t)$ and $\lambda_L(t)$ appear as spiral curves on the $\lambda_G$ and $\lambda_L$ sheets, respectively.

The nature of the corresponding instantaneous eigenvectors $|\phi_G(t)\rangle$ and $|\phi_L(t)\rangle$ are understood to represent amplifying (gain) and attenuating (loss) modes, respectively. We note at the start and end points ($t = 0$ and $T$) that $\xi = 0$ and the instantaneous eigenvectors $|\phi_G\rangle$ and $|\phi_L\rangle$ are either an even mode, $|\text{even}\rangle = 2^{-1/2} [1\ 1]^T$, with the eigenvalue $\lambda = +1$ or an odd mode, $|\text{odd}\rangle = 2^{-1/2} [1 -1]^T$, with $\lambda = -1$. In particular, $|\phi_G(0)\rangle = |\phi_L(0)\rangle = |\text{even}\rangle$ whereas $|\phi_L(T)\rangle = |\phi_G(T)\rangle = |\text{odd}\rangle$. Therefore, $|\phi_G(t:0 \to T)\rangle$ and $|\phi_L(t:0 \to T)\rangle$ continuously evolve from $|\text{even}\rangle$ into $|\text{odd}\rangle$. These mode swapping evolution passages of the instantaneous eigenvectors around the PT-symmetric EP is often referred to as an adiabatic state flip. The adiabatic state flip and associated geometric-phase accumulation were experimentally confirmed in coupled microwave cavities[18,19] and exciton–polaritonic quantum billiard experiments[20]. These adiabatic

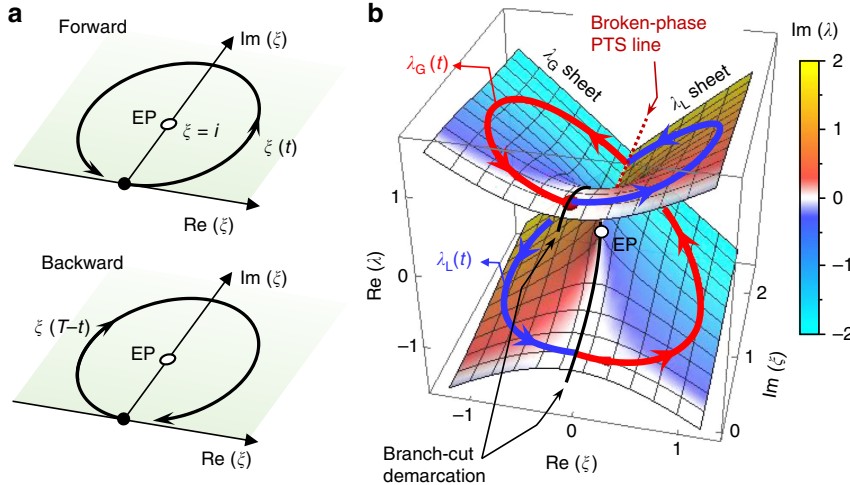

**Figure 2 | Optical eigensystem characteristics of the unidirectional converter region.** (**a**) Complex $\xi$ trajectories for the forward (upper) and backward (lower) propagation cases. (**b**) Complex eigenvalue spectra and instantaneous eigenvalue trajectories on the complex $\xi$ space. The wireframe surface and surface skin colour indicate real and imaginary parts, respectively. The instantaneous eigenvalue trajectories are calculated for the circular $\xi(t)$ trajectory in **a**. Arrows on the trajectory curves indicate the direction of the parametric change with time $t$.

interaction properties have been confirmed by tracking the instantaneous eigenstates in the quasi-stationary limit. The complete dynamics of state evolution in non-Hermitian systems with significant imaginary eigenvalue splitting in general involve highly non-adiabatic behaviour associated with an anti-adiabatic state jump occurring under appropriate initial-state and parametric conditions[21–24]. The anti-adiabatic state jump is a key interaction leading to the time-asymmetric state-evolution passages in our proposed system.

The essence of the anti-adiabatic state jump and associated time-asymmetric effects is revealed in a case where the initial state is given by either one of the instantaneous eigenvectors, that is, $|\psi(0)\rangle = |\phi_\mu(0)\rangle$. In the limit $\langle \phi_\mu^* | \psi \rangle \gg \langle \phi_v^* | \psi \rangle$ that is likely for a slowly varying system satisfying the well-known quantum adiabatic condition[22] $\langle \partial \mathbf{H}/\partial t \rangle \ll |\lambda_\mu - \lambda_v|^2$, the probability amplitude for non-adiabatic transition from $|\phi_\mu\rangle$ to $|\phi_v\rangle$ is approximated by:

$$C_{v\mu}(t) = \frac{\langle \phi_v^* | \psi \rangle}{\langle \phi_\mu^* | \psi \rangle} \approx \int_0^t g_{v\mu}(\tau) \exp\left[i(\Lambda_v - \Lambda_\mu)(t - \tau)\right] d\tau, \quad (5)$$

where $g_{v\mu}(t) = \langle \phi_v^* | d\mathbf{H}/dt | \phi_\mu \rangle / (\lambda_v - \lambda_\mu)$ is a non-adiabatic coupling constant and $\Lambda_\mu = (t - \tau)^{-1} \int_\tau^t \lambda_\mu(t') dt'$ is average eigenvalue over the time domain $[\tau, t]$. Note here that the inner product $\langle \cdot^* | \cdot \rangle$ of two state vectors is the $c$-product following the biorthogonal treatment for non-Hermitian systems[25,26]. See Supplementary Note 1 for the detailed derivation. For Hermitian Hamiltonians that essentially involve purely real eigenvalues $\lambda_\mu$ and $\lambda_v$, $C_{v\mu}(t)$ remains negligible for slowly varying $\mathbf{H}(t)$ under the quantum adiabatic condition. Consequently, the time evolution of the state $|\psi(t)\rangle$ in a Hermitian system simply follows the instantaneous eigenvector passage $|\phi_\mu(t)\rangle$.

However, in non-Hermitian cases where the eigenvalues include significant imaginary parts, the standard quantum adiabatic theorem fails to properly describe the state evolution. A radical breakdown of the standard quantum adiabatic theorem for the forward propagation state $|\psi_{fw}(t)\rangle$ is seen in Fig. 3a as an anti-adiabatic jump of the expectation value $\langle \mathbf{H}(t) \rangle_{fw} \equiv \langle \psi_{fw}^* | \mathbf{H}(t) | \psi_{fw} \rangle / \langle \psi_{fw}^* | \psi_{fw} \rangle$ from the $\lambda_L$ sheet to the $\lambda_G$ sheet, in stark contrast to the highly adiabatic expectation

value passage $\langle \mathbf{H}(t) \rangle_{bw} \equiv \langle \psi_{bw}^* | \mathbf{H}(t) | \psi_{bw} \rangle / \langle \psi_{bw}^* | \psi_{bw} \rangle$ for the backward state in Fig. 3b. Therein, the forward expectation value $\langle \mathbf{H}(t) \rangle_{fw}$ passage under the initial condition $\langle \mathbf{H}(0) \rangle_{fw} = +1$ follows the $\lambda_L$ sheet in the beginning, undergoes the anti-adiabatic jump towards the $\lambda_G$ sheet and eventually ends up at $\langle \mathbf{H}(T) \rangle_{fw} = +1$. This forward expectation value passage indicates the sequential mode transition of $|even\rangle \rightarrow |\phi_L(t)\rangle \rightarrow |\phi_G(T-t)\rangle \rightarrow |even\rangle$ involving the anti-adiabatic jump from $|\phi_L\rangle$ to $|\phi_G\rangle$. In the numerical calculation of Fig. 1c, manifestation of the anti-adiabatic jump appears as a transition of the modal intensity profile from an initially attenuating pattern to an amplifying pattern in the region indicated by a white dotted box. Considering general aspects of the anti-adiabatic jump, equation (5) implies that any evolution passage on the $\lambda_L$ sheet undergoes the anti-adiabatic jump towards the $\lambda_G$ sheet whenever the dwell time on the $\lambda_L$ sheet exceeds a critical time interval $T_c \equiv \text{Im}(\Lambda_L - \Lambda_G)^{-1}$. A conceptually equivalent time parameter to $T_c$ was noted by Milburn et al.[27] as a time of stability loss delay for small-radius encircling-an-EP parametric paths.

In the backward propagation case, the expectation value $\langle \mathbf{H}(t) \rangle_{bw}$ passage indicates a mode-swapping adiabatic evolution that follows a simple process of $|even\rangle \rightarrow |\phi_G(t)\rangle \rightarrow |odd\rangle$ as shown in Fig. 3b. In particular, $|\psi_{bw}(t)\rangle$ follows the instantaneous eigenvector $|\phi_G(t)\rangle$ for $\mathbf{H}(t)$ quickly varying in time even beyond the standard quantum adiabatic condition. This type of super-adiabatic evolution passage is expected for any dynamic state under the condition $|\psi\rangle \approx |\phi_G\rangle$ for which the expectation value $\langle \mathbf{H} \rangle$ appears on the $\lambda_G$ sheet. In Supplementary Note 1, we explain in greater detail the fundamental reasons for the anti-adiabatic jump and the super-adiabatic evolution passages, depending on the direction of time-evolution and the initial conditions.

Looking at the time-domain profiles of the channel amplitude ratio $A_2/A_1$, the dynamic properties of the state evolution associated with the forward anti-adiabatic and backward super-adiabatic passages are more clearly observable. We show the real and imaginary parts of $A_2/A_1$ for $|\psi_{fw}\rangle$ and $|\psi_{bw}\rangle$ as functions of $t$ in Fig. 3c. The forward passage $|\psi_{fw}(t)\rangle$ undergoes the following sequential behaviour from $t = 0$ to $t = T = 25$: it is launched with $A_2/A_1 = 1$ corresponding to state $|even\rangle$; slowly deviates with a spiral oscillation from $|\phi_L\rangle$ to $t \approx 4$; suddenly diverges from $|\phi_L\rangle$ at $t \approx T_c \approx 6$, indicating the anti-adiabatic jump; converges to $|\phi_G\rangle$

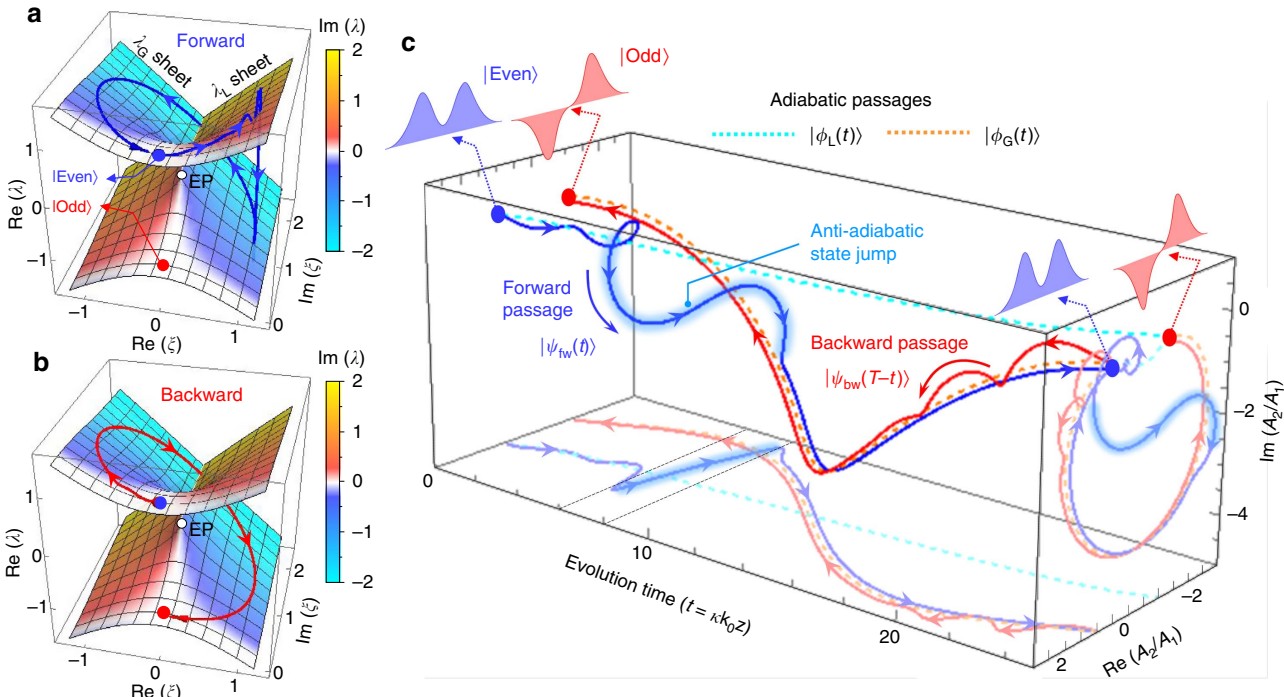

**Figure 3 | Time-asymmetric evolution properties in the unidirectional converter region.** (**a**,**b**) Energy expectation value $\langle H \rangle$ trajectories for the forward (**a**) and backward (**b**) propagation cases, respectively. (**c**) Time evolution of relative channel amplitudes $A_2/A_1$ for the forward and backward propagation cases in comparison with purely adiabatic evolution passages. In stark contrast to the backward passage (red curve) following the adiabatic passage that switches the mode's transversal symmetry, the forward passage undergoes an anti-adiabatic state jump (the region highlighted in light blue) and consequently the final state maintains the even-transversal symmetry. In (**c**), projections of the dynamic and adiabatic passages onto the $t$-Re$(A_2/A_1)$ and Re$(A_2/A_1)$-Im$(A_2/A_1)$ planes are provided for better understanding of the time-domain response. The fictitious time domain of the anti-adiabatic jump for the forward passage is indicated by dashed line on the $t$-Re$(A_2/A_1)$ plane. In this analysis, we calculate $|\psi_{\mathrm{fw}}(t)\rangle$, $|\psi_{\mathrm{bw}}(t)\rangle$ and $\langle H \rangle$ using the classical Runge–Kutta (RK4) method with total evolution-time parameter $T = 25$.

at $t \approx 10$; and finally ends up in state $|\mathrm{even}\rangle$ through the exact adiabatic $|\phi_G\rangle$ passage. In contrast, the backward passage of $|\psi_{\mathrm{bw}}(T-t)\rangle$ from the initial state $|\mathrm{even}\rangle$ simply follows the adiabatic passage of $|\phi_G\rangle$ to eventually end up at state $|\mathrm{odd}\rangle$. Interestingly, a spiral oscillation of $|\psi_{\mathrm{bw}}(T-t)\rangle$ because of non-adiabatic coupling over the domain of $20 \leq T-t \leq 25$ vanishes within $T_c \approx 6$, indicating the super-adiabatic evolution property.

Anti-adiabatic properties and associated time-asymmetric effects have been studied previously. The inevitability of time-asymmetric anti-adiabatic jumps in non-Hermitian systems has been theoretically argued in the case of molecular vibrational state transfer because of chirped pulses[22,23] and for dual-mode optical waveguides[28]. Very recently, this seemingly counterintuitive effect was experimentally verified independently in a microwave channel waveguide structure[29] and in an optomechanical system[30]. In previous studies, the circular geometry for parametric trajectories was not the unique case and arbitrary geometrical paths enclosing an EP could be used for inducing the required time-asymmetric anti-adiabatic jump.

The time asymmetry in the state-evolution passages provides a key principle for our proposed nonreciprocal device concept. This property by itself is not yet sufficient to accomplish strictly nonreciprocal photonic transmission as the fundamental Lorentz reciprocity theorem for linear electromagnetism in stationary optical systems requires[15,16]. Hence, we introduce a gain saturation nonlinearity to break the strict reciprocity in transmission. In Fig. 4a,b, we show even-mode power ratio $P_{\mathrm{sym}}/P_{\mathrm{tot}}$ profiles for the forward and backward passages in the linear ($I_S = \infty$) and nonlinear ($I_S = I_0$) cases, respectively.

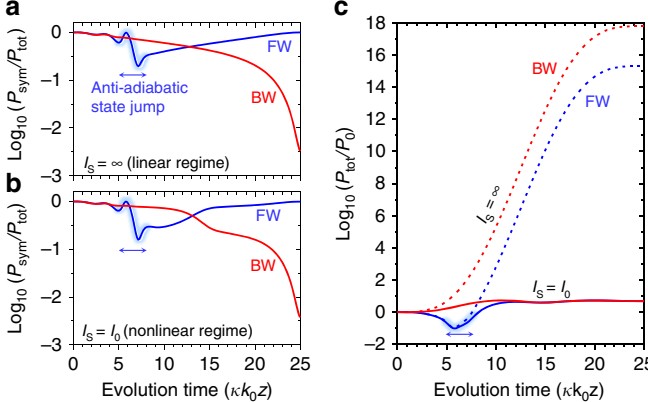

**Figure 4 | Strictly nonreciprocal transmission with gain saturation.** (**a**,**b**) Even-mode power ratio $P_{\mathrm{sym}}/P_{\mathrm{tot}}$ profiles while encircling an EP for the forward (FW, blue curve) and backward (BW, red curve) passages with no gain saturation ($I_S = \infty$) and strong gain saturation ($I_S = I_0$), respectively. Here, $I_S$ denotes a gain saturation intensity constant. (**c**) Total power history while encircling an EP for the forward (blue curves) and backward (red curves) passages with no gain saturation ($I_S = \infty$, dotted curves) and strong gain saturation ($I_S = I_0$, solid curves). Regions of anti-adiabatic state jumps are indicated by light-blue highlighting and a double-sided arrow.

Therein, we find negligible differences in the time-asymmetric state-evolution properties between the linear and nonlinear cases. The key properties such as the parity-preserving forward

evolution (unity order $P_{sym}/P_{tot}$ value at $t = 25$) with an anti-adiabatic state-jump signature at $t \approx T_c \approx 6$, and the parity-exchanging, adiabatic backward evolution (low $P_{sym}/P_{tot}$ value at $t = 25$) persist for the highly nonlinear case in the almost identical manner to those for the linear case. In a quantitative comparison, the differences in the final $P_{sym}/P_{tot}$ values between the linear and nonlinear cases are below 0.02%. The role of the gain saturation nonlinearity is to utilize the time-asymmetric even-mode power ratio as the nonreciprocal transmission ratio. In Fig. 4c, we show the forward and backward total power $P_{tot}(t)$ profiles in the linear ($I_0 << I_S = \infty$) and nonlinear ($I_0 = I_S$) cases. In the final state at $t = 25$, the forward and backward $P_{tot}$ values in the nonlinear case are identical to each other at $P_{tot} = 4.8 P_0$ ($\log_{10}(P_{tot}/P_0) = 0.681$). This is because the gain saturation nonlinearity equalizes the total power once the modal intensity level in the amplifying arm becomes significant with respect to the saturation level of $I_S$. Assuming that only the even mode contributes to the final transmission toward the output single-mode waveguide, the nonreciprocal transmission ratio is identical to the ratio of the forward even-mode power ratio to the backward even-mode power ratio.

In the linear case, however, the final forward $P_{tot}$ value is substantially lower than the final backward $P_{tot}$ value as shown in Fig. 4c, implying that the time-asymmetric even-mode power ratio does not yield a significant nonreciprocity. In particular, $P_{tot}$ for the backward passage $|\psi_{bw}\rangle$ grows monotonically as the state simply follows the amplifying-mode $|\phi_G(T-t)\rangle$ passage. In contrast, $P_{tot}$ for the forward passage $|\psi_{fw}\rangle$ is strongly attenuated for its transient dwell time in the attenuating-state $|\phi_L(t)\rangle$ passage before the anti-adiabatic jump occurs at $t \approx T_c \approx 6$. This difference in the power-amplification history results in the observed difference in $P_{tot}$ depending on the propagation direction. Essentially, the subtle processes of non-adiabatic coupling, amplification and transient attenuation in the linear regime render the partial even-mode powers $P_{sym}$ for $|\psi_{bw}\rangle$ and $|\psi_{fw}\rangle$ at $t = T$ exactly identical to each other. Therefore, although the time-asymmetric state-evolution passages induce large difference in the even-mode power ratio $P_{sym}/P_{tot}$ for the two opposite encircling-an-EP directions, the reciprocity in the transmission is unbroken in the linear case.

As discussed earlier for the strong gain saturation regime, in which the total power equalization effect is fully realized, the NTR is given by the ratio of the forward even-mode power ratio to the backward even-mode power ratio. Following this argument, under the condition of $|\langle even|\psi_{fw}(T)\rangle|^2 \approx |\langle \psi_{fw}(T)|\psi_{fw}(T)\rangle|^2 \approx |\langle \psi_{bw}(T)|\psi_{bw}(T)\rangle|^2$, the NTR takes on a simple form

$$R_{NTR} \approx \left| \frac{\langle even | \psi_{fw}(T)\rangle}{\langle even | \psi_{bw}(T)\rangle} \right|^2 \approx |C_{LG}(T)|^{-2} \approx \alpha T^2 \quad (6)$$

as approximated from equation (5) for $T \gg 1$. Here, the coefficient $\alpha$ is determined by the geometry of $\xi(t)$ on the complex $\xi$ plane. In equation (5) for $t = T$, a partial integration domain capturing a significant contribution of the instantaneous non-adiabatic coupling amplitude $g_{LG}(\tau)$ to $C_{LG}(T)$ corresponds to $T - T_c < \tau < T$. Taking this effective domain for the integration in equation (5), an approximate expression is found such that $|C_{LG}(T)| \approx |f(T)| \cdot T_c/T$, where $f = (\partial \xi/\partial \zeta) \cdot \langle \phi_L|\partial \mathbf{H}/\partial \xi|\phi_G\rangle \cdot (\lambda_L - \lambda_G)^{-1}$ with $\zeta = z/L$. Importantly, equation (6) implies that the NTR is determined by the purity of the backward dynamic state $|\psi_{bw}\rangle$ with respect to the amplifying eigenvector passage $|\phi_G\rangle$. The NTR is not affected by the phase difference between the eigenmodes or by interference-induced power beating effects that are highly sensitive to the operating wavelength and device length in general. In Fig. 5, we show the dependence of $R_{NTR}$ on $T$ for

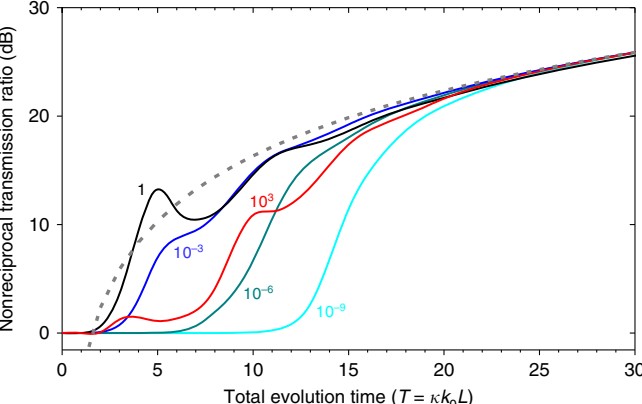

**Figure 5 | Dependence of the nonreciprocal transmission ratio on the total evolution-time parameter T.** Each curve shows the NTR for a given incident intensity to saturation intensity ratio $I_0/I_S$ as indicated by a value in the same colour as the corresponding curve. The grey dashed curve represents an $\alpha T^2$ curve fit following equation (6).

several values of $I_0/I_S$. The $T^2$ dependence of the NTR at large $T$ is confirmed quantitatively. As implied in equation (6), the $R_{NTR}(T)$ profiles have no periodic feature that would normally be associated with interference-induced power beating on the scale of the conventional beat length $\Delta T = 1$, that is, $\Delta L = (\kappa k_0)^{-1}$. This property is a unique feature of our proposed concept enabling broadband optical nonreciprocity.

We evaluate the spectral characteristics of the NTR and forward transmission efficiency (FTE) as major performance parameters. In the spectral analysis, we recall the model device used in Fig. 1c–e and calculate the NTR and FTE spectra for $L = 1$, 5 and 10 mm, whereas other structural parameters and optical constants remain identical to those indicated in the caption of Fig. 1. We use the classical Runge–Kutta (RK4) method for solving the nonlinear coupled-mode model of equations (1–3). The results are given in Fig. 6. Quantitative agreement between the spectral curves obtained from the nonlinear coupled-mode model and the symbols obtained from the fully vectorial finite-element method confirms the validity of our theoretical approach. Major performance parameters estimated from the data in Fig. 6 are summarized in Table 1. When compared with the 1-to-100 GHz $\Delta v_{10\text{-dB}}$ of resonator-based optical isolators[3,4,31] and the 1 THz $\Delta v_{10\text{-dB}}$ of dynamic refractive index modulation approaches[1,2], the bandwidths over which our NTR remains high ($> 10$ dB) and our FTE near unity are remarkably high, exceeding 100 THz.

The essential underlying mechanisms for the strong non-reciprocal property are the EP-induced asymmetry in the even-mode power ratio and the total power equalization effect because of the gain saturation nonlinearity. These two effects have different spectral properties, leading to characteristic spectral features such as a bell-shaped profile in the NTR and a threshold-like behaviour in the FTE near the wavelength of 1.18 μm, as observed in Fig. 6. First, according to the argument leading to equation (6), the EP-induced asymmetry in the even-mode power ratio is a quadratic function of the total evolution-time parameter $T = \kappa k_0 L$ that is monotonically increasing with wavelength because $\kappa$ exponentially grows with wavelength as determined by a field overlap between the guided modes in waveguides 1 and 2. In contrast, the gain saturation-induced total power equalization effect becomes weaker in the longer wavelength domain as the parametric $\xi(t) = \Delta n_1/\kappa$ trajectory in the shorter wavelength range enters deeper into the highly non-Hermitian domain above

the EP where large imaginary eigenvalue splitting results in a rapid power amplification of the gain mode $|\phi_G(t)\rangle$ passage—see the $\xi(t)$ trajectory with respect to the EP on top of Fig. 6. Therefore, as determined by the tradeoff between the increasing EP-induced asymmetry in the even-mode power ratio and decreasing gain saturation-induced total power equalization effect with wavelength, the spectral region of maximum NTR appears in the intermediate wavelength region where the $\xi(t)$ trajectory closely encircles the EP, as numerically confirmed in Fig. 6a. In the short wavelength limit, near the wavelength of 0.6 μm where the $\xi(t)$ trajectory is far away from the EP during the whole evolution passage, the EP-induced time asymmetry is not significant because of small $T$, yielding a small NTR. On the other hand, in the long wavelength limit, near the wavelength of 1.6 μm, the gain saturation-induced total power equalization effect is weak and the optical transmission becomes reciprocal

regardless of the degree of time asymmetry in the even-mode power ratio.

In addition, the imaginary eigenvalue splitting has a threshold-like behaviour near the EP, and the circular $\xi(t)$ trajectory does not encircle the EP any more at wavelengths longer than 1.18 μm. This behaviour is responsible for the threshold-like character in the FTE spectra in Fig. 6b. Interestingly, the fairly high NTR persists in the longer wavelength region where the circular $\xi(t)$ trajectory excludes the EP. In this region, forward propagation involves two anti-adiabatic jumps preventing symmetry exchange, whereas backward propagation undergoes a single anti-adiabatic jump resulting in symmetry exchange from the even to odd mode. Explaining the spectral characteristics of the FTE, a main consideration is to derive a relation between the maximum output intensity and key factors such as gain saturation intensity, gain/loss coefficients and the parametric geometry of the $\xi(t)$ trajectory. Although this problem is beyond the scope of this paper, it would provide important information for systematically optimizing the device for applications.

## Discussion

Considering the experimental feasibility of the proposed nonreciprocal device concept, an important issue is to determine efficient approaches to synthesize the required complex effective index profiles to dynamically encircle an EP. Although various methods such as position-dependent doping or the deposition of gain and loss agents on appropriately-coupled channel wave-guides can be considered, we briefly introduce a lithographic approach that does not involve yet-unestablished fabrication issues.

The approach is illustrated in Fig. 7. The unidirectional mode converter region consists of two coupled-channel waveguides with an adjacent side patch. In this design, the real effective index profile is created by a $z$-dependent waveguide-core width $w_2(z)$, whereas the imaginary effective index profiles are created via the $z$-dependent gap-width $d_s(z)$ between the waveguide core and side patch. In particular, the side patch induces leakage radiation from the guided mode toward the adjacent side patch at a desired rate for a given $w_2(z)$ value, thereby controlling the imaginary effective index in an independent manner. In Supplementary Fig. 1 and Supplementary Note 2, we present an example design based on dye-doped polymer waveguides. Therein, the optimized design produces the desired encircling-an-EP parametric modal evolution with the essential attributes. Advantageously, the example design has geometrical parameters that are highly favourable for standard nanolithographic fabrication processes and yields a theoretical performance consistent with the two-dimensional design assumed for Fig. 6.

Additional discussions of the operating bandwidth restrictions because of the gain bandwidth of the selected emitter species and a Kramers–Kronig relation between the real and imaginary indices are also provided in Supplementary Note 2. Importantly, the proposed design maintains the required complex effective index profiles even if the optical constants of the constituent materials drift, thereby ensuring a more stable performance than

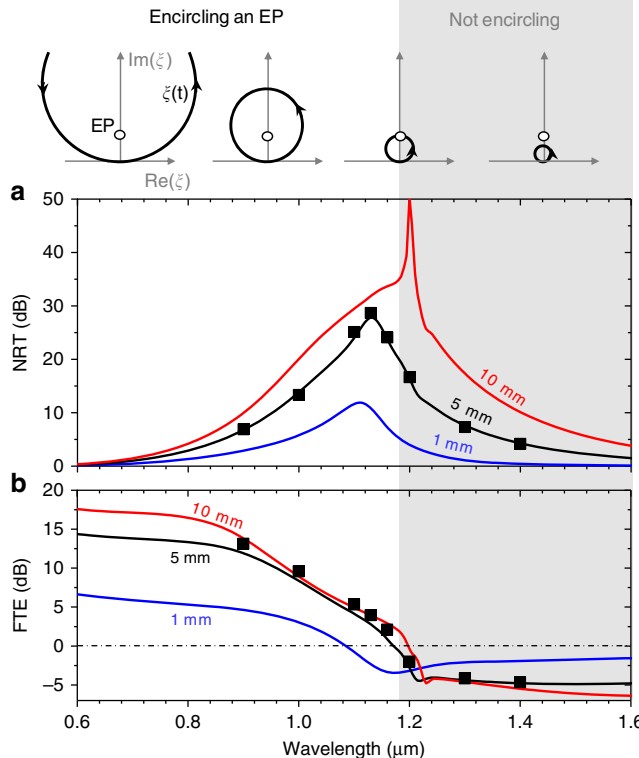

**Figure 6 | Broadband performance characteristics. (a)** Spectral profiles of the nonreciprocal transmission ratio (NTR) for several device length values of $L = 1$ mm (blue), 5 mm (black) and 10 mm (red). **(b)** Spectral profiles of the forward transmission efficiency (FTE) for $L = 1$, 5 and 10 mm. The curves in both panels are obtained by solving the nonlinear coupled-mode model of equations (1–3), whereas the square symbols for $L = 5$ mm are calculated by the finite-element method. We assume $I_0/I_S = 1$ in this analysis. The top diagrams above (**a**) schematically show a wavelength-dependent $\xi(t)$ trajectory with respect to the EP.

**Table 1 | Major performance parameters for model devices.**

| $L$ | $[R_{NTR}]_{max}$ | $\lambda_{max}$ | $\Delta\lambda_{\text{10-dB}}$ | $\Delta\nu_{\text{10-dB}}$ | $\langle FTE \rangle_{avg}$ |
|---|---|---|---|---|---|
| 1 mm | 11.9 dB | 1.11 μm | 62 nm | 15.2 THz | 0.977 |
| 5 mm | 27.8 dB | 1.13 μm | 305 nm | 76.5 THz | 1.081 |
| 10 mm | 34.8 dB | 1.18 μm | 500 nm | 119.0 THz | 1.072 |

The listed parameters are peak nonreciprocal transmission ratio (NTR) $[R_{NTR}]_{max}$, peak NTR wavelength $\lambda_{max}$, 10 dB NTR bandwidth $\Delta\lambda_{\text{10 dB}}$ and $\Delta\nu_{\text{10 dB}}$ and average forward transmission efficiency $\langle FTE \rangle_{avg}$ over the 10 dB NTR band.

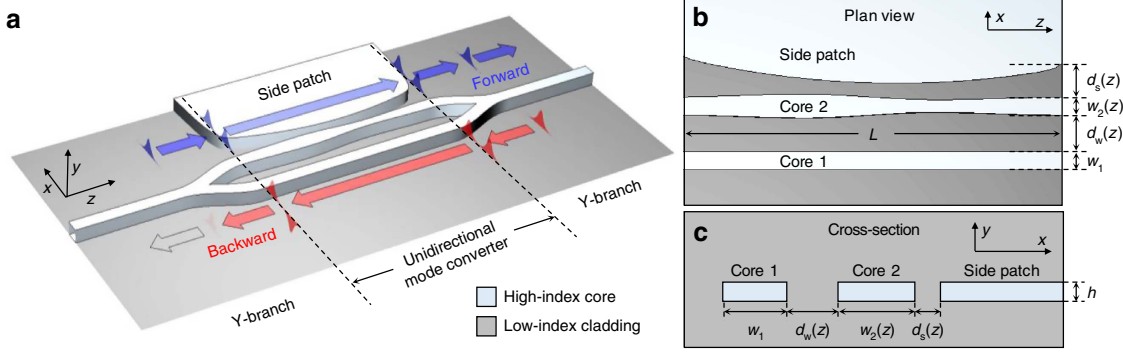

**Figure 7 | Schematic of a proposed design based on lithographically generated complex effective index modulation.** Device structure (**a**) in three-dimensional (3D) view, (**b**) in plan view in the $x$–$z$ plane and (**c**) in cross-section in the $x$–$y$ plane. In this design, the appropriately modulated inter-waveguide spacing $d_w(z)$, waveguide width $w_2(z)$ and waveguide side-patch spacing $d_s(z)$ create real and imaginary effective index profiles identical to those indicated in Fig. 1b.

other potential approaches based on position-dependent doping of gain and loss agents.

In summary, we proposed a principle for on-chip broadband optical non-reciprocity enabled by nonlinear EP dynamics. Employing judiciously interrelated optical gain and absorption distributions, the proposed device architecture and associated operating principles produce high-quality on-chip optical nonreciprocity over a spectral bandwidth exceeding 100 THz. The anticipated performance is notably distinctive from previous approaches.

Thus, it is of great interest to experimentally realize the proposed device idea utilizing available materials and nanophotonic structures. In optimizing practical designs, various geometries producing parametric paths that encircle an EP should be taken into account to maximize the operating bandwidth, NTR and FTE. Limitations in the experimental performance should be carefully investigated in consideration of power thresholds and bandwidth restrictions resulting from the chosen optical gain mechanism as well as fabrication imperfections. In addition, we note that strong optical confinement and field enhancement in nanoplasmonic and high-index semiconductor platforms may yield much smaller device footprints and a lower power threshold. In a broader perspective, we hope that our results stimulate extensive research on various non-Hermitian optical effects and associated device applications.

**Data availability**. The data that support the findings of this study are available from the corresponding authors on request.

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

## Acknowledgements

This research was supported in part by the Basic Science Research Program (NRF-2015R1A2A2A01007553) and by the Global Frontier Program through the National Research Foundation (NRF) of Korea funded by the Ministry of Science, ICT & Future Planning (NRF-2014M3A6B3063708).

## Author contributions

Y.C., C.H. and S.H.S. conceived the original concept and initiated the work. Y.C. and J.W.Y. developed the theory and model. C.H. and Y.C. performed numerical analyses. J.W.Y. and Y.C. organized the results. J.W.Y., P.B., Y.C. and S.H.S. wrote the manuscript. All authors discussed the results. Y.C. and C.H. contributed equally to this work.

## Additional information

**Competing financial interests:** The authors declare no competing financial interests.

**Publisher's note**: 

