## [Peer Review File · Nature Communications]

Reviewers' comments:

Reviewer #1 (Remarks to the Author):

In the paper by Youngsun Choi et al. the authors propose a non-resonant non-reciprocal device based on a coupled waveguide containing distributed gain and loss. Due to the non-trivial evolution of states in such a non-Hermitian system, the forward and backward modes undergo highly asymmetric evolution in the waveguide. When this asymmetry is combined with a non-linear response the device shows a large degree of non-reciprocity, with almost zero backwards transmission.

Overall I think the article is very well written and contains some interesting results. However, I had concerns about (a) the novelty of the results, and (b) the precise details of the proposed experiment. Due to these concerns I cannot recommend publication in Nature Communications.

(a) Concerning novelty: As the authors discuss, it is well known that when a state is slowly transported around a non-Hermitian degeneracy then it can undergo highly non-adiabatic transitions (one particular reference I don't think was included - Uzdin and Moiseyev Phys. Rev. A 031804 2012). Therefore Figs. 2 and 3 mostly serve to demonstrate this for this particular waveguide system.

It is also known that combining non-Hermitian physics with non-linearity can lead to non-reciprocal transmission (for the case of pt symmetry, one missing reference is Ramezani, Phys Rev. A 043803 2010). In this respect, the results of figure 1 do not seem very different from figure 1 of the aforementioned reference (perhaps I am missing something). Therefore, despite the fact that this work is of interest and seems to be correct, I am left wondering what the new insight of this paper is (i.e. what warrants publication in Nature Communications rather than elsewhere?).

(b) Concerning the experimental realization: The paper is concerned with the theory and numerical modelling of the system, rather than its experimental realization. I think it aims to be one step closer to an experiment, and some emphasis is placed on the precise value of the system parameters and the potential bandwidth of the device. However, it assumes the ability to achieve the index profile given in (1-2). This seems to be experimentally challenging, and to potentially limit the bandwidth of the device (doesn't this profile have to obey the Kramers-Kronig relations?). Yet there is barely any comment regarding the synthesis of this profile (or any similar one).

Reviewer #2 (Remarks to the Author):

In this paper the authors basically suggest an experimental setup to detect the asymmetric effect discussed by Moiseyev and his coworkers in series of works (Ref 18,19,20 that should be in the focus of their paper. Maybe even mention in the abstract). They also show how the 2 by 2 results are well reproduced in numerical simulations of the WG propagation.

They did something in a spirit of what in Refs.19,20 they were trying to do some time ago: making a non-reciprocal device by using nonlinearity (intensity-dependence) of the refraction index, and a loop around EP in parameter space.

Without getting too deep in their theoretical arguments, according to numerics it looks all right. So, in general, I would be in favor of this paper.

One thing I still not understand (as the major shortcoming).

Most of the work they repeat previous results on EP encircling. However, when it comes to the nonlinear effect in Fig.4b, which is their central point, the role of EP remains unclear. What Fig.4b shows is that the effect of nonlinearity makes everything almost opposite to what happened in the linear case (Fig.4a).

I could not find any theoretical justification in the paper explaining why this happens.

Another side of this question: if the nonlinear behavior is so different, is it true that the EP encircling was important at all in this device?

Some part of the explanation seems to be related to Fig.5 (encircling/not encircling) but it is not sufficient. After revision the paper should be reconsidered for publication in Nature Communication

PS Detail theoretical explanation of the effect of branch points (exceptional points in the spectrum of non-Hermitian Hamiltonian) is given in Chapter 9 of the book on Non-Hermitian Quantum Mechanics, published by Cambridge University Press at 2011.

Reviewer #3 (Remarks to the Author):

The authors of the submitted manuscript propose the principle for on-chip optical non-reciprocity exploiting the singularity related to the exceptional point in the eigenfrequency spectrum of non-Hermitian systems. On the base of performed theoretical analysis supported by modeling results, they claim the possibility to achieve an extremely broadband nonreciprocal performance over 100 THz, while maintaining a forward-to-backward transmission ratio exceeding 10 dB, and high forward transmission efficiency (~100%).

The publication of this manuscript is very timely since encircling exceptional in non-Hermitian systems is a hot topic. The submitted work represents probably the very first proposal of a realistic device exploiting this concept. Its publication would be highly beneficial to stimulate the research oriented on the practical implementation of such advanced theoretical concept into the real life devices. Overall, the manuscript is for the most part well written, and certainly of interest for the Parity-Time symmetry and optics communities. I therefore recommend its publication. Below are some minor issues to be addressed by the authors.

1) On page 2 line 38 it can be read: "nonreciprocal transmission is obtained as opposed to strictly reciprocal PT-symmetric effects in the passive case".

In fact the correct statement would be rather "reciprocal PT-symmetric effects below exceptional point".

2) Though it would be in principle possible to determine from the supplied in the manuscript data the linear gain (dB/cm) in the example considered by the authors, it would be better to explicitly provide this value in order to help the reader to evaluate the requirements for the practical implementation of such a device. Furthermore, it would be also highly useful providing the total level of amplification for a given refractive index profile of an isolated waveguide. Same suggestion concerning attenuation level applies also for the lossy waveguide.

3) To highlight the occurrence of a non-adiabatic jump, the reviewer suggest to provide corresponding comments to the evolution of light intensity shown in Fig 1(c).

Reviewer #1 (Remarks to the Author + Author Response)

In the paper by Youngsun Choi et al. the authors propose a non-resonant non-reciprocal device based on a coupled waveguide containing distributed gain and loss. Due to the non-trivial evolution of states in such a non-Hermitian system, the forward and backward modes undergo highly asymmetric evolution in the waveguide. When this asymmetry is combined with a non-linear response the device shows a large degree of non-reciprocity, with almost zero backwards transmission.

Overall I think the article is very well written and contains some interesting results. However, I had concerns about (a) the novelty of the results, and (b) the precise details of the proposed experiment. Due to these concerns I cannot recommend publication in Nature Communications.

Reviewer Comment #1) (a) Concerning novelty: As the authors discuss, it is well known that when a state is slowly transported around a non-Hermitian degeneracy then it can undergo highly non-adiabatic transitions (one particular reference I don't think was included - Uzdin and Moiseyev Phys. Rev. A 031804 2012). Therefore Figs. 2 and 3 mostly serve to demonstrate this for this particular waveguide system.

It is also known that combining non-Hermitian physics with non-linearity can lead to non-reciprocal transmission (for the case of pt symmetry, one missing reference is Ramezani, Phys Rev. A 043803 2010). In this respect, the results of figure 1 do not seem very different from figure 1 of the aforementioned reference (perhaps I am missing something). Therefore, despite the fact that this work is of interest and seems to be correct, I am left wondering what the new insight of this paper is (i.e. what warrants publication in Nature Communications rather than elsewhere?).

Author Response #1) Responding to this criticism, we would remind Reviewer #1 that the purpose of our paper is to show possibility of a dynamically encircling-an-EP interaction for realizing yet-illusory on-chip broadband optical nonreciprocity for the first time, not just incrementally putting the previously known concept on a readily expectable application area. Hence, in the original manuscript, we include *a new method how the time-asymmetric state jump property that is essentially reciprocal in the linear interaction regime can make a plausible optical-nonlinear device architecture highly nonreciprocal over a broad spectral domain*. As pointed out, the time-asymmetric evolution in non-Hermitian systems is indeed a theoretically well-known effect. However, its feasibility to produce a broadband optical nonreciprocity has never been reported so far. The nonreciprocal transmission in a nonlinear PT-symmetric coupled-waveguide system was theoretically treated in [Ramezani *et al.*, Phys. Rev. A **82**, 043803 (2010)] as Reviewer #1 mentions in this comment. However, the key effect in this reference paper is an enhanced nonreciprocity due to the spontaneous PT-symmetry breaking, not the time-asymmetric evolution effect during dynamically encircling an EP. Therefore, the underlying physics in our paper is totally different from those previously reported including this reference paper. Please note that our proposed device structure happens to be PT-symmetric at the center of the device whereas the anti-adiabatic state jump can occur in any highly non-Hermitian domain which is not necessarily PT-symmetric. In addition, none of the previous reports has suggested the use of nonlinear, non-Hermitian physics for obtaining the *broadband optical nonreciprocity in integration-compatible photonic architectures*. In this manuscript, we spotlight this aspect for the first time in the optical physics and engineering communities.

Reviewer Comment #2) (b) Concerning the experimental realization: The paper is concerned with the theory and numerical modelling of the system, rather than its experimental realization. I think it aims to be one step closer to an experiment, and some emphasis is placed on the precise value of the system parameters and the potential bandwidth of the device. However, it assumes the ability to achieve the index profile given in (1-2). This seems to be experimentally challenging, and to potentially limit the bandwidth of the device (doesn't this profile have to obey the Kramers-Kronig relations?). Yet there is barely any comment regarding the synthesis of this profile (or any similar one).

Author Response #2) We thank Reviewer #1 for this comment. We agree that the experimental possibility is a very important aspect. Hence, we have included additional supplementary information as **Supplementary Figure 1** and **Supplementary Note 2** where we provide substantial further analysis results considering a more realistic device

design operating under our proposed principle. Therein, the required complex effective-index profiles are created using a single lithographic patterning procedure. Importantly, this approach does not involve any yet-unestablished fabrication issues and favorably avoids potential complication due to a Kramers-Kronig relation between the real and imaginary index-modulation profiles. Explaining the approach and implications of the results provided by the additional supplementary information, **Fig. 7** has been added and associated text description is provided at the beginning of Discussion section on page 10 in the main text as provided below:

Figure 7 | Proposed design based on lithographically-generated complex effective-index modulation. a-c, Schematic drawing of a design in (a) 3D view , (b) plan view in the x - z plane, and (c) cross-section in the x - y plane.

[Line 241 on page 10] “Considering the experimental feasibility of the proposed nonreciprocal device concept, an important issue is to determine efficient approaches to synthesize the required complex effective-index profiles to dynamically encircle an EP. While various methods such as position-dependent doping or the deposition of gain and loss agents on appropriately-coupled channel waveguides can be considered, we briefly introduce a lithographic approach that does not involve yet-unestablished fabrication issues.

The approach is illustrated in Fig. 7. The unidirectional mode converter region consists of two coupled-channel waveguides with an adjacent side patch. In this design, the real effective index profile is created by a z -dependent waveguide-core width $w_2(z)$, while the imaginary effective index profiles are created via the z -dependent gap-width $d_s(z)$ between the waveguide core and side patch. In particular, the side patch induces leakage radiation from the guided mode toward the adjacent side patch at a desired rate for a given $w_2(z)$ value, thereby controlling the imaginary effective index in an independent manner. In Supplementary Fig. 1 and Supplementary Note 2, we present an example design based on dye-doped polymer waveguides. Therein, the optimized design produces the desired encircling-an-EP parametric modal evolution with the essential attributes. Advantageously, the example design has geometrical parameters that are highly favorable for standard nano-lithographic fabrication processes and yields a theoretical performance consistent with the 2D design assumed for Fig. 6.

Additional discussions of the operating bandwidth restrictions due to the gain bandwidth of the selected emitter species and a Kramers-Kronig relation between the real and imaginary indices are also provided in Supplementary Note 2. Importantly, the proposed design maintains the required complex effective index profiles even if the optical constants of the constituent materials drift , thereby ensuring a more stable performance than other potential approaches based on position-dependent doping of gain and loss agents.”

Reviewer #2 (Remarks to the Author + Author Response)

In this paper the authors basically suggest an experimental setup to detect the asymmetric effect discussed by Moiseyev and his coworkers in series of works (Ref 18,19,20 that should be in the focus of their paper. Maybe even mention in the abstract). They also show how the 2 by 2 results are well reproduced in numerical simulations of the WG propagation.

They did something in a spirit of what in Refs.19, 20 they were trying to do some time ago: making a non-reciprocal device by using nonlinearity (intensity-dependence) of the refraction index, and a loop around EP in parameter space.

Without getting too deep in their theoretical arguments, according to numerics it looks all right. So, in general, I would be in favor of this paper.

Reviewer Comment #3) One thing I still not understand (as the major shortcoming):

Most of the work they repeat previous results on EP encircling. However, when it comes to the nonlinear effect in Fig.4b, which is their central point, the role of EP remains unclear. What Fig.4b shows is that the effect of nonlinearity makes everything almost opposite to what happened in the linear case (Fig.4a). I could not find any theoretical justification in the paper explaining why this happens.

Another side of this question: if the nonlinear behavior is so different, is it true that the EP encircling was important at all in this device? Some part of the explanation seems to be related to Fig.5 (encircling/not encircling) but it is not sufficient. After revision the paper should be reconsidered for publication in Nature Communication.

Author Response #3) The role of an encircling-an-EP evolution is to induce large difference in the ratio of the even-mode power to the total power between the two opposite propagation directions. Independently, the role of the gain-saturation nonlinearity is to equalize the total powers for the two opposite propagation directions. In the original manuscript, we tried to make this point as clear as possible with Fig. 4 and associated text description. However, the explanations in the text was revealed in the data shown in Fig. 4 not in an explicit manner. Therefore, we have revised Fig. 4 and associated text in a completely new form as attached below:

Figure 4 | Strictly nonreciprocal transmission with gain saturation. **a** and **b**, Even-mode power ratio P_{sym}/P_{tot} profiles while encircling an EP for the forward (FW, blue curve) and backward (BW, red curve) passages with no gain saturation ($I_S = \infty$) and strong gain saturation ($I_S = I_0$), respectively. **c**, Total power history while encircling an EP for the forward (blue curves) and backward (red curves) passages with no gain saturation ($I_S = \infty$, dotted curves) and strong gain saturation ($I_S = I_0$, solid curves). Regions of anti-adiabatic state jumps are indicated by light-blue highlighting and a double-sided arrow.

[Line 167 on page 7] “The time-asymmetry in the state evolution passages provides a key principle for our proposed nonreciprocal device concept. This property by itself is not yet sufficient to accomplish strictly nonreciprocal photonic transmission as the fundamental Lorentz reciprocity theorem for linear electromagnetism in stationary optical systems restricts^{15,16}. Hence, we introduce a gain-saturation nonlinearity to break the *strict reciprocity* in transmission. In Figs. 4a and 4b, we show even-mode power ratio P_{sym}/P_{tot} profiles for the forward and backward passages in the linear ($I_S = \infty$) and nonlinear ($I_S = I_0$) cases, respectively. Therein, we find negligible differences in the time-asymmetric state-evolution

properties between the linear and nonlinear cases. The key properties such as the parity-preserving forward evolution (unity-order $P_{\text{sym}}/P_{\text{tot}}$ value at $t = 25$) with an anti-adiabatic state-jump signature at $t \approx T_c \approx 6$, and the parity-exchanging, adiabatic backward evolution (low $P_{\text{sym}}/P_{\text{tot}}$ value at $t = 25$) persist for the highly nonlinear case in the almost identical manners to those for the linear case. In a quantitative comparison, the differences in the final $P_{\text{sym}}/P_{\text{tot}}$ values between the linear and nonlinear cases are below 0.02%. The role of the gain-saturation nonlinearity is to utilize the time-asymmetric even-mode power ratio as the nonreciprocal transmission ratio. In Fig. 4c, we show the forward and backward total power $P_{\text{tot}}(t)$ profiles in the linear ($I_0 \ll I_S = \infty$) and nonlinear ($I_0 = I_S$) cases. In the final state at $t = 25$, the forward and backward P_{tot} values in the nonlinear case are identical to each other at $P_{\text{tot}} = 4.8P_0$ ($\log_{10}(P_{\text{tot}}/P_0) = 0.681$). This is because the gain-saturation nonlinearity equalizes the total power once the modal intensity level in the amplifying arm becomes significant with respect to the saturation level of I_S . Assuming that only the even mode contributes to the final transmission toward the output single-mode waveguide, the nonreciprocal transmission ratio is identical to the ratio of the forward even-mode power ratio to the backward even-mode power ratio.

In the linear case, however, the final forward P_{tot} value is substantially lower than the final backward P_{tot} value as shown in Fig. 4c, implying that the time-asymmetric even-mode power ratio does not yield a significant nonreciprocity. In particular, P_{tot} for the backward passage $|\psi_{\text{bw}}\rangle$ grows monotonically as the state simply follows the amplifying-mode $|\phi_G(T-t)\rangle$ passage. In contrast, P_{tot} for the forward passage $|\psi_{\text{fw}}\rangle$ is strongly attenuated for its transient dwell time in the attenuating-state $|\phi_e(t)\rangle$ passage before the anti-adiabatic jump occurs at $t \approx T_c \approx 6$. This difference in the power-amplification history results in the observed difference in P_{tot} depending on the propagation direction. Essentially, the subtle processes of non-adiabatic coupling, amplification, and transient attenuation in the linear regime render the partial even-mode powers P_{sym} for $|\psi_{\text{bw}}\rangle$ and $|\psi_{\text{fw}}\rangle$ at $t = T$ exactly identical to each other. Therefore, although the time-asymmetric state-evolution passages induce large difference in the even-mode power ratio $P_{\text{sym}}/P_{\text{tot}}$ for the two opposite encircling-an-EP directions, the reciprocity in the transmission is unbroken in the linear case.

As discussed earlier for the strong gain-saturation regime, in which the total-power equalization effect is fully realized, the NTR is given by the ratio of the forward even-mode power ratio to the backward even-mode power ratio. Following this argument, under the condition of $|\langle \text{even} | \psi_{\text{fw}}(T) \rangle|^2 \approx |\langle \psi_{\text{fw}}(T) | \psi_{\text{fw}}(T) \rangle|^2 \approx |\langle \psi_{\text{bw}}(T) | \psi_{\text{bw}}(T) \rangle|^2$, the NTR takes on a simple form

$$R_{\text{NTR}} \approx \frac{|\langle \text{even} | \psi_{\text{fw}}(T) \rangle|^2}{|\langle \text{even} | \psi_{\text{bw}}(T) \rangle|^2} \approx |C_{\text{LG}}(T)|^{-2} \approx \alpha T^2 \quad (5)$$

as approximated from Eq. (4) for $T \gg 1$.³¹ Here, the coefficient α is determined by the geometry of $\xi(t)$ on the complex ξ plane. Importantly, Eq. (5) implies that the NTR is determined by the purity of the backward dynamic state $|\psi_{\text{bw}}\rangle$ with respect to the amplifying eigenvector passage $|\phi_G\rangle$. The NTR is not affected by the phase difference between the eigenmodes nor by interference-induced power beating effects which are highly sensitive to the operating wavelength and device length in general. In Fig. 5, we show the dependence of R_{NTR} on T for several values of I_0/I_S . The T^2 dependence of the NTR at large T is confirmed quantitatively. As implied in Eq. (5), the $R_{\text{NTR}}(T)$ profiles have no periodic feature that would normally be associated with interference-induced power beating on the scale of the conventional beat length $\Delta T = 1$, i.e., $\Delta L = (\kappa k_0)^{-1}$. This property is a unique feature of our proposed concept enabling broadband optical nonreciprocity.”

In Fig. 4 and the text description after revision, the separate roles of an encircling-an-EP evolution and the gain-saturation nonlinearity are clearly revealed, also confirming that the effect of EP is obviously operative in the nonlinear case. Associated with this change, Fig. 4c in the original manuscript has been separated as Fig. 5 in the revised manuscript. In addition, explanations regarding Fig. 5 in the original manuscript (it is Fig. 6 in the revised manuscript) has been rewritten so that the observed spectral features are understood based on the spectral dependence of the encircling-an-EP properties and the nonlinear power-equalization effect as attached below:

[Line 216 on page 9] “The essential underlying mechanisms for the strong nonreciprocal property are the EP-induced asymmetry in the even-mode power ratio and the total-power equalization effect due to the gain-saturation nonlinearity. These two effects have different spectral properties, leading to characteristic spectral features such as a bell-shaped

profile in the NTR and a threshold-like behavior in the FTE near the wavelength of $1.18 \mu\text{m}$, as observed in Fig. 6. First, according to the argument leading to Eq. (5), the EP-induced asymmetry in the even-mode power ratio is a quadratic function of the total evolution-time parameter $T = \kappa k_0 L$ which is monotonically increasing with wavelength because κ exponentially grows with wavelength as determined by a field overlap between the guided modes in waveguides 1 and 2. In contrast, the gain-saturation-induced total power equalization effect becomes weaker in the longer wavelength domain as the parametric $\zeta(t) (= \Delta n_1 / \kappa)$ trajectory in the shorter wavelength range enters deeper into the highly non-Hermitian domain above the EP where large imaginary-eigenvalue splitting results in a rapid power amplification of the gain-mode $|\phi_G(t)\rangle$ passage - see the $\zeta(t) (= \Delta n_1 / \kappa)$ trajectory with respect to the EP on top of Fig. 6. Therefore, as determined by the trade-off between the increasing EP-induced asymmetry in the even-mode power ratio and decreasing gain-saturation-induced total-power equalization effect with wavelength, the spectral region of maximum NTR appears in the intermediate wavelength region where the $\zeta(t)$ trajectory closely encircles the EP, as numerically confirmed in Fig. 6a. In the short wavelength limit, near the wavelength of $0.6 \mu\text{m}$ where the $\zeta(t)$ trajectory is far away from the EP during the whole evolution passage, the EP-induced time asymmetry is not significant due to small T , yielding a small NTR. On the other hand, in the long wavelength limit, near the wavelength of $1.6 \mu\text{m}$, the gain-saturation-induced total power equalization effect is weak and the optical transmission becomes reciprocal regardless of the degree of time asymmetry in the even-mode power ratio.

In addition, the imaginary-eigenvalue splitting has a threshold-like behavior near the EP, and the circular $\zeta(t)$ trajectory does not encircle the EP any more at wavelengths longer than $1.18 \mu\text{m}$. This behavior is responsible for the threshold-like character in the FTE spectra in Fig. 6b. Interestingly, the fairly high NTR persists in the longer wavelength region where the circular $\zeta(t)$ trajectory excludes the EP. In this region, forward propagation involves two anti-adiabatic jumps preventing symmetry exchange, while backward propagation undergoes a single anti-adiabatic jump resulting in symmetry exchange from the even to odd mode."

Reviewer Comment #4) PS Detail theoretical explanation of the effect of branch points (exceptional points in the spectrum of non-Hermitian Hamiltonian) is given in Chapter 9 of the book on Non-Hermitian Quantum Mechanics, published by Cambridge University Press at 2011.

Author Response #4) We have included this book as Ref. [17] with an added sentence as shown below:

[Line 92 on page 4] "The parametric spectra of eigenvalues $\lambda_{\pm} = \pm(1+\zeta^2)^{1/2}$ of \mathbf{H} on the complex ζ plane have characteristic singularities at $\zeta = \pm i$ corresponding to a pair of PT-symmetric EPs. A comprehensive description and the general features of non-Hermitian singularities of this kind are found in [17]."

Reviewer #3 (Remarks to the Author + Author Response):

The authors of the submitted manuscript propose the principle for on-chip optical non-reciprocity exploiting the singularity related to the exceptional point in the eigenfrequency spectrum of non-Hermitian systems. On the base of performed theoretical analysis supported by modeling results, they claim the possibility to achieve an extremely broadband nonreciprocal performance over 100 THz, while maintaining a forward-to-backward transmission ratio exceeding 10 dB, and high forward transmission efficiency (~100%).

The publication of this manuscript is very timely since encircling exceptional in non-Hermitian systems is a hot topic. The submitted work represents probably the very first proposal of a realistic device exploiting this concept. Its publication would be highly beneficial to stimulate the research oriented on the practical implementation of such advanced theoretical concept into the real life devices. Overall, the manuscript is for the most part well written, and certainly of interest for the Parity-Time symmetry and optics communities. I therefore recommend its publication. Below are some minor issues to be addressed by the authors.

Reviewer Comment #5) 1) On page 2 line 38 it can be read: "nonreciprocal transmission is obtained as opposed to strictly reciprocal PT-symmetric effects in the passive case". In fact the correct statement would be rather "reciprocal PT-symmetric effects below exceptional point".

Author Response #5) Even in the broken-PT-symmetry phase, the optical transmission should be reciprocal if there is no nonlinear effect or time-varying components included in the system. This fact was confirmed in [Fan *et al.* Comment on "Nonreciprocal light propagation in a silicon photonic circuit". *Science* 335, 38 (2012)] and [Jalas *et al.* What is – and what is not – an optical isolator. *Nat. Photon.* 7, 579-582 (2013)]. Moreover, it is completely consistent with what we are showing in Fig. 4. Thereby, these reference papers has been included as [15] and [16], and the relevant sentence has been revised as shown below:

[Line 36 on page 2] "Including auxiliary nonlinear effects, such as the optical Kerr effect and gain saturation,^{6,7} strong nonreciprocal transmission is obtained as opposed to strictly reciprocal PT-symmetric effects in the linear and stationary systems⁹⁻¹⁶."

Reviewer Comment #6) 2) Though it would be in principle possible to determine from the supplied in the manuscript data the linear gain (dB/cm) in the example considered by the authors, it would be better to explicitly provide this value in order to help the reader to evaluate the requirements for the practical implementation of such a device. Furthermore, it would be also highly useful providing the total level of amplification for a given refractive index profile of an isolated waveguide. Same suggestion concerning attenuation level applies also for the lossy waveguide.

Author Response #6) In the revised manuscript, we provide the requested information due to this reviewer comment as shown below:

[Line 70 on page 3] "Figures 1c to 1e summarize 2-D FEM (Finite-Element Method) calculations assuming single-mode slab waveguides with $\Delta n_0 = 4.26 \times 10^{-4}$, $L = 5$ mm, a waveguide core width of 1 μm , a core separation distance of 2 μm , and an operating wavelength of 1.13 μm (other parameters and conditions are given in the figure caption). For an isolated waveguide of these parameters, the maximum linear modal gain/attenuation constants are ± 411.4 $\text{dB} \cdot \text{cm}^{-1}$ and the total amplification/attenuation over the 5-mm-long unidirectional converter region are ± 102.8 dB. These levels of optical gain and loss are readily obtainable with variety of optical gain materials such as conventional direct-bandgap semiconductors and dye-doped polymers."

Reviewer Comment #7) 3) To highlight the occurrence of a non-adiabatic jump, the reviewer suggest to provide corresponding comments to the evolution of light intensity shown in Fig 1(c).

Author Response #7) In the revised manuscript, we have indicated the region of the anti-adiabatic jump by a white-dotted box in Fig. 1c and have stated with an added sentence:

[Line 138 on page 6] "In the numerical calculation of Fig. 1c, manifestation of the anti-adiabatic jump appears as a transition of the modal intensity profile from an initially attenuating pattern to an amplifying pattern in the region indicated by a white-dotted box."

Reviewers' comments:

Reviewer #1 (Remarks to the Author):

The authors have responded clearly to both of my concerns. I am more convinced by the synthesis of the profiles which - from the new discussion given in the supplementary material (SM) - look to be feasible and probably retain some reasonable bandwidth. Some further quantitative analysis of the application of the Kramers-Kronig relations beyond the text at the end of the SM would have been even more convincing, but this can be analysed in future work.

As for the novelty, I was originally reading the work looking for novel physics, which the authors agree is not the purpose of this manuscript. As the authors emphasise, the novelty is in the design and the functionality of this proposed device, which will be of interest to those working on the engineering of optical devices. With the addition of the new SM the proposed device looks feasible.

I therefore recommend publication of the new version of this manuscript.

Reviewer #2 (Remarks to the Author):

The abstract of the Letter is misleading and leaves the wrong impression that this work represents a novel unknown phenomenon of time asymmetric dynamics.

The authors should clarify in the new version of the abstract that the idea of time asymmetric "switches" is known and first presented in their Ref 21 and first demonstrated in experiments in Nature 537, 80-83 (2016) and Nature 537, 76-79 (2016) and then add an explanation on the new information in their work (as for example "first showing how the time-asymmetric state jump property that is essentially reciprocal in the linear interaction regime can make a plausible optical-nonlinear device architecture highly nonreciprocal over a broad spectral domain").

Reviewer #3 (Remarks to the Author):

The reviewer considers that in the revised version authors have taken into account the recommended suggestions.

There is however one point that attracted reviewer attention and merits to be clarified. The results displayed in Fig. 5 are showing that when the fictitious time parameter $T > 20$, the nonreciprocal transmission ratio tends to the asymptote given by $\alpha \cdot T^2$ and doesn't depend of the initial conditions determined by I_0 . Could the authors verify that this is not due to the dissipative properties of the system meaning that the input signal is lost and the recovered output signal is just amplified spontaneous emission noise.

The reviewer would be in favor of the publication of this paper provided that the authors dissipate the mentioned above worry.

Author response to the review report

Reviewer #1

The authors have responded clearly to both of my concerns. I am more convinced by the synthesis of the profiles which - from the new discussion given in the supplementary material (SM) - look to be feasible and probably retain some reasonable bandwidth. Some further quantitative analysis of the application of the Kramers-Kronig relations beyond the text at the end of the SM would have been even more convincing, but this can be analysed in future work.

Author Response) We thank Reviewer #1 for his/her comments. In addition, we agree that the consequences of the Kramers-Kronig relation in our proposed nonreciprocal interaction are definitely worthy of further in-depth study.

Reviewer #2

The abstract of the Letter is misleading and leaves the wrong impression that this work represents a novel unknown phenomenon of time asymmetric dynamics.

The authors should clarify in the new version of the abstract that the idea of time asymmetric “switches” is known and first presented in their Ref 21 and first demonstrated in experiments in Nature 537, 80-83 (2016) and Nature 537, 76–79 (2016) and then add an explanation on the new information in their work (as for example “first showing how the time-asymmetric state jump property that is essentially reciprocal in the linear interaction regime can make a plausible optical-nonlinear device architecture highly nonreciprocal over a broad spectral domain”).

Author Response) We have revised the abstract following the reviewer’s comments to more clearly state the main result of our paper. The new abstract is shown below:

Time-asymmetric state-evolution properties while encircling an exceptional point (EP) are presently of great interest in search of new principles for controlling atomic and optical systems. Here, we show that encircling-an-EP interactions that are essentially reciprocal in the linear interaction regime make a plausible nonlinear integrated-optical device architecture highly nonreciprocal over an extremely broad spectrum. In the proposed strategy, we describe an experimentally-realizable coupled-waveguide structure that supports an encircling-an-EP parametric evolution under the influence of a gain-saturation nonlinearity. Using an intuitive time-dependent Hamiltonian and rigorous numerical computations, we demonstrate strictly nonreciprocal optical transmission with a forward-to-backward transmission ratio exceeding 10 dB and high forward transmission efficiency (~100%) persisting over an extremely broad bandwidth approaching 100 THz. This predicted performance is unprecedented and strongly encourages experimental realization of the proposed concept to establish a practical on-chip optical non-reciprocal element for ultra-short laser pulses and broadband high-density optical signal processing.

In addition, the mentioned reference papers [Nature 537, 80-83 (2016)] and [Nature 537, 76-79 (2016)] are included as [31] and [32] respectively with an additional statement:

Very recently, this seemingly counter-intuitive effect was experimentally verified independently in a microwave channel waveguide structure³⁰ and in an optomechanical system³¹.

in line 165 on page 7.

Reviewer #3

The reviewer considers that in the revised version authors have taken into account the recommended suggestions. There is however one point that attracted reviewer attention and merits to be clarified. The results displayed in Fig. 5 are showing that when the fictitious time parameter $T > 20$, the nonreciprocal transmission ratio tends to the asymptote given by αT^2 and doesn't depend of the initial conditions determined by I_0 . Could the authors verify that this is not due to the dissipative properties of the system meaning that the input signal is lost and the recovered output signal is just amplified spontaneous emission noise.

The reviewer would be in favor of the publication of this paper provided that the authors dissipate the mentioned above worry.

Author Response) We thank the reviewer for this comment. Amplified spontaneous emission processes are unrelated to the results reported in our paper because our theoretical analyses assume a coherent input optical mode as the seed that interacts with the gain medium (which is non-dissipative). A significant part of our manuscript is devoted to explain the asymptotic behavior in the NTR found in Fig. 5 as originating directly from the gain-saturation nonlinearity and subsequent power regulation effect when the signal intensity reaches a level comparable to the saturation intensity constant I_s (see text descriptions on pages 7 to 8 for this matter).

Also, as we consider possible experiments using available laser sources, we do not expect significant consequences due to amplified spontaneous-emission noise. Following the standard theory of optical emission processes, the ratio η of the stimulation emission (StE) rate to the spontaneous emission (SpE) rate is determined by the relation $\eta = u\lambda^3(4h)^{-1}$, where u , λ and h denote spectral power density, vacuum wavelength of the injected optical signal, and Planck's constant, respectively. Assuming an incident coherent mode with a mode-field cross-sectional area of $10 \mu\text{m}^2$, optical power of 1 mW, and spectral linewidth of 10 MHz at a vacuum wavelength of $1 \mu\text{m}$, which are consistent with our analyzed cases, the StE-to-SpE ratio η has a value $\sim 10^8 \gg 1$.

In addition, our structure does not include any feedback for creating a self-sustained optical oscillation triggered by spontaneous emission.

In summary, although amplified spontaneous emission exists in a gain medium in general, we do not think that it would play an important role in our device concept, so we prefer not to add any discussion along these lines for fear of creating confusion. Therefore, we have no change to our manuscript following this comment by Reviewer #3.

REVIEWERS' COMMENTS:

Reviewer #3 (Remarks to the Author):

The authors have provided answer to the reviewer concern. Some further analysis bringing for instance the relation between the maximal output intensity as function of saturation intensity, gain and loss coefficients in the system would be highly welcome, but this problem can be treated in some future work.

Reviewer #3

The authors have provided answer to the reviewer concern. Some further analysis bringing for instance the relation between the maximal output intensity as function of saturation intensity, gain and loss coefficients in the system would be highly welcome, but this problem can be treated in some future work.

Author Response) We agree that further analysis on the output intensity as a function of saturation intensity, gain and loss coefficient is definitely an important problem to be treated in a follow-up publication. Therefore, we have added a sentence describing this point in line 246 on page 10. It is attached below.

“Explaining the spectral characteristics of the FTE, a main consideration is to derive a relation between the maximum output intensity and key factors such as gain-saturation intensity, gain/loss coefficients, and the parametric geometry of the $\zeta(t)$ trajectory. Although this problem is beyond the scope of this paper, it would provide important information for systematically optimizing the device for applications.”